# Fully Automated Segmentation of the Left Ventricle in Small Animal Cardiac MRI

**Hao Xu**
Department of Engineering Science
University of Oxford
hao.xu@eng.ox.ac.uk

**Jurgen E. Schneider**
Institute of Cardiovascular and Metabolic Medicine
University of Leeds
j.e.schneider@leeds.ac.uk

**Alistair A. Young**
Department of Anatomy with Radiology
University of Auckland
a.young@auckland.ac.nz

**Vicente Grau**
Department of Engineering Science
University of Oxford
vicente.grau@eng.ox.ac.uk

## Abstract

The study of cardiovascular diseases requires viable animal models, and small animals have become models of choice due to their significant advantages. Segmentation from cine-MR image has become the gold standard for cardiac function assessment. While many image analysis methods have been developed for clinical studies, similar techniques are generally lacking for preclinical cases. Recent application of neural networks has shown encouraging results in several medical imaging applications. For cardiac cine-MR image segmentation, convolutional neural networks and recurrent neural networks have been successfully used for segmentation of the left ventricle. However, most methods only use a stack of short axis images, which introduces inaccuracy and uncertainty for cardiac function assessment in 3D, especially considering the existence of misalignment between slices. In this paper, we propose an efficient and 3D consistent segmentation method for small animal cardiac MR images taking advantage of a combination of long-axis and short-axis images, by combining convolutional neural networks and the guide-point modelling method. Unlike in most clinical studies, we also focus on training with small datasets, as is common in preclinical studies, and show accurate results with only 12 cine-MR sequences.

## 1 Introduction

The study of cardiovascular disease requires suitable animal models, and in the last half a century using small-animal models to study cardiovascular pathophysiology has shown its great value [14]. Particularly, genetically altered mouse models provide unique approaches to understanding the mechanisms and progress of heart failure, and there are important practical reasons for using mice, such as the genetic similarity with humans and the dramatic reduction in cost comparing with large animals (e.g. pig). Due to the different image acquisition approaches for mice and human hearts, cardiac MR scans also vary in quality. Human cardiac MR images tend to have larger Signal-to-Noise Ratio (SNR) and clear intensity features around the myocardium, which provides benefits for image processing. Even though the segmentation of mice cardiac MR images is a more challenging task, the number of studies published up to date is much smaller, e.g. [22].

In recent years, machine learning methods, and particularly the application of Fully Convolutional Neural Networks (FCN) [5], have shown improvements on the segmentation results from conventional techniques for both the Left Ventricle (LV) and the Right Ventricle (RV) [1, 9, 10, 18], thanks in

1st Conference on Medical Imaging with Deep Learning (MIDL 2018), Amsterdam, The Netherlands.

part to the development of large publicly available databases (e.g. the Cardiac Atlas Project [2]) for training the models. For most publicly available databases, each dataset consists of a stack of Short-Axis (SAX) images, and therefore the focus of cardiac MR image segmentation has been the application of 2D FCNs on SAX images. Cardiac MR datasets also include routinely images acquired along the Long-Axis direction (LAX), but segmentation studies incorporating these are much less common.

One issue associated with 2D FCNs is that some poor segmentations have conflicts with prior information shown in [18], where examples are depicted in Figure 1. The red contours in the images are segmented endocardial contours and the green contours are segmented epicardial contours. For some of these cases endocardial contours and epicardial contours intersect, and for some other cases endocardial contours are bigger than epicardial contours. Incorporating prior information into the segmentation method has shown some advantage in producing more accurate and plausible results [11]. Another issue is caused by the combination of cardiac motion and breathing motion, leading to misalignment between slices. Recurrent FCNs (RFCN) [15] and 3D FCNs [13] have been applied to reduce the effect of these motion artefacts. For both cases, the number of parameters of the neural network are increased dramatically comparing to popular 2D FCNs, requiring more datasets to avoid overfitting, and the LAX images are not used for segmentation in any of these.

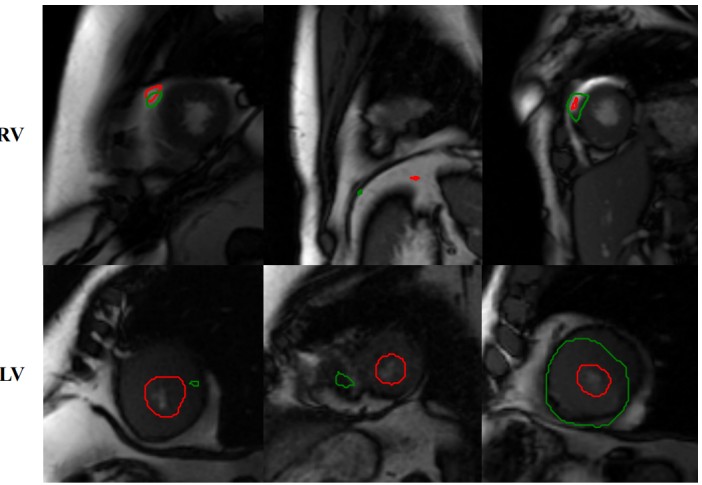

Figure 1: Poor segmentation examples from [18].

The main reason that LAX images are not widely used in FCN segmentation methods, even though they provide significant additional information to SAX images, is the lack of labelled datasets. This is not required in other applications such as super-resolution. A multi-channel Convolutional Neural Network (CNN), which uses a combination of LAX and SAX images as inputs, has been proposed to generate high resolution cardiac MR images from low resolution ones [12], where the intensity features of the intersecting region from one slice are transformed to the corresponding region on another slice. Compared to RFCN and Anatomically Constrained Neural Networks (ACNN), a multi-channel CNN only duplicates part of the network and thus the increase of the number of parameters is much smaller, i.e. more suitable for training neural networks with a small number of datasets.

Unlike the emerging machine learning algorithms, Guide-Point Modelling (GPM) is an established, clinically accepted method for the generation of the LV shape based on a 3D finite-element model. GPM has been used in several studies and is currently in use in clinical environments as part of the commercial software Cardiac Image Modelling (CIM), which has been licensed to the largest MRI manufacturer in the world (Siemens) and distributed into hospitals internationally [6]. It has also been used to generate an LV shape atlas [7] using the Cardiac Atlas Project database [2].

In this paper, we propose a fully automated segmentation method for cine-MR images in mice models, combining a multi-layer input U-Net [16] (which has an even smaller number of parameters comparing with multi-channel architecture for deep networks) for 2D segmentation and GPM for 3D model generation. To our best knowledge, this is the first Deep Learning (DL) method for LV

segmentation which combines information from both LAX and SAX images. To improve the 3D consistency of the segmentation results, we fuse information from intersecting slices in the U-Net and incorporate the GPM method to produce 3D LV surfaces from the 2D segmentations. The design of the U-Net and the inclusion of an established segmentation method (GPM) take into consideration the typically small database that we address. Our method is a fully automated pipeline which takes a collection of cine-MR images of intersected slices (a stack of SAX images and two LAX images are provided, but any combinations of intersecting slices are compatible) and gives a complete set of 2D and 3D segmentation results which are then used to identify key frames and evaluate cardiac functions.

## 2 Method

Our segmentation method consists of two main components: a variation of U-Net and GPM. In the following sections, we will first describe these two parts separately, and then introduce the complete segmentation method.

### 2.1 U-Net for 2D Segmentation

We propose a multi-stage segmentation method with two U-Nets, which take a stack of SAX images (N parallel slices) and two LAX images as input. We name the two U-Nets Image-Net (I-Net) and Transformation-Net (T-Net), where I-Net only takes image intensity as input, and T-Net, which takes both the intensity of the slice to be segmented and the transformed segmentation of other intersecting slices as input, respectively. The output of the network consists in two probability maps arranged in two layers: one corresponding to the LV cavity (the region enclosed by the endocardial contour) and one to the LV cavity plus the LV myocardium (the region enclosed by epicardial contour). The network architectures are shown in Figure 2.

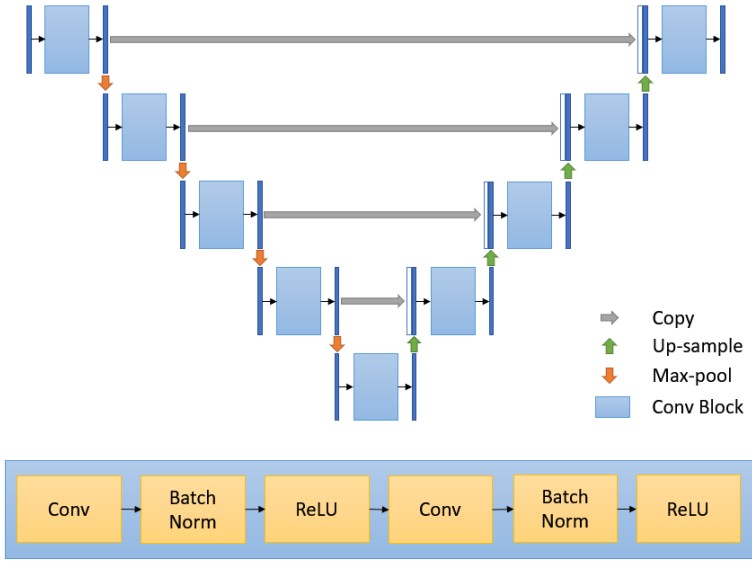

Figure 2: Network architecture and convolutional block.

From the implementation point of view, the only difference between the two neural networks is that the input layer of I-Net has the shape of $128 \times 128 \times 1$ and the input layer of T-Net has the shape of $128 \times 128 \times 3$, where the additional two layers are the transformation layers as explained later in this section. Small filters with a size of $3 \times 3$ are used for all convolutional layers, inspired by the VGG network [17]. Also by following the philosophy of popular architectures such as VGG, ResNet [3] and U-Net, we set the number of filters with the following rules: for layers with the same resolution we use the same number of features; we double the number of features after pooling; we halve the number of features after up-sampling. Padding was also applied for patch size preservation. We use 64 filters for the first convolutional block, and include four pooling layers with stride of 2 in

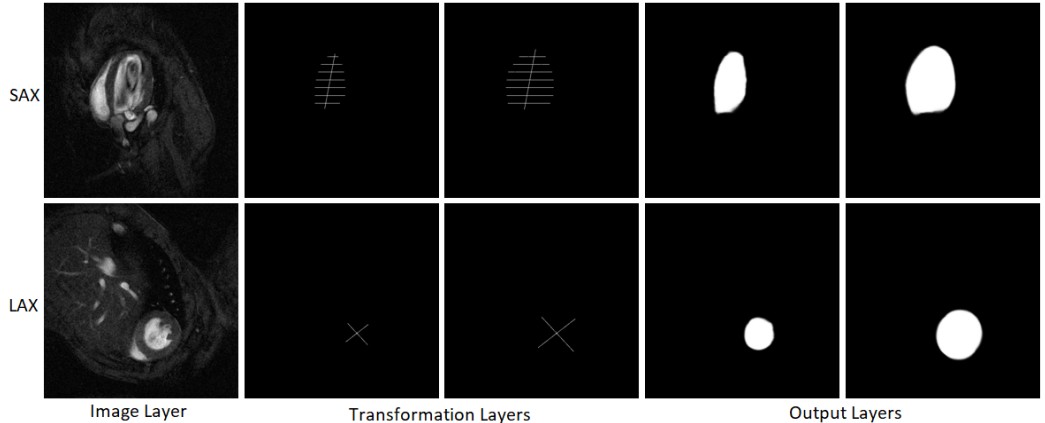

Figure 3: Input and output layers examples.

the network. Before and after each pooling and up-sampling layer, there is a convolutional block, consisting of two sets of convolutional layers followed by a batch normalization layer [4] and ReLU activation [8] in a sequence. Examples of the input-/output-layers of the U-Nets are shown in Figure 3.

## 2.2  3D Model Generation

The GPM method is applied to generate a 3D model of the LV, including endocardial and epicardial surfaces. GPM takes contour point coordinates as input. We applied a threshold at 0.5 to the probability map from the U-Net to generate a binary mask, where the coordinates of the boundary points are used as input to GPM. GPM as a stand-alone process works in the following way: first, fiducial landmarks (two points for defining the apex and base, and two points for defining the mitral valve) are selected manually from the LAX images to define a prolate spheroidal coordinate system [19] and to form the initial model by aligning and scaling an average generic model to the defined coordinate system [20].

In our automated approach, the fiducial landmarks are approximated by using the most basal SAX contours. After the formation of the initial model, which is a tetrahedral closed mesh forming the endocardial and epicardial surfaces, the contour points from U-Net results are then fitted by linear least squares to construct the model in the prolate spheroidal coordinate system:

$$x = f \cosh(\lambda) \cos(\mu) \tag{1}$$
$$y = f \sinh(\lambda) \sin(\mu) \cos(\theta) \tag{2}$$
$$z = f \sinh(\lambda) \sin(\mu) \sin(\theta) \tag{3}$$

where $(\lambda, \mu, \theta)$ are the radial, longitudinal, and circumferential coordinates of the prolate spheroidal coordinate system and $(x, y, z)$ are the corresponding Cartesian coordinates. An objective function $E$ is then defined to be used in a minimization process:

$$E = \omega S(\lambda) + \sum_{g \in C} \lambda_g^2 \tag{4}$$

where $S(\lambda)$ is the Sobolev smoothing term, $\omega$ is the smoothing weight, and $\lambda_g$ is the displacement from the initial solution to the contour data point in element coordinates. In the above equation, the first term serves as a regulariser, forcing the values of $\lambda$ to vary smoothly within the mesh (and thus favouring meshes that resemble a prolate spheroid), while the second term tries to fit the mesh to the contour points. The summation spans across the available contour points from all the slices ($g \in C$) other than the points above the basal plane [20].

The manual approach usually employed with GPM involves additional steps to generate more accurate and plausible 3D models, including in-plane transformation of images for misalignment correction,

and adding manual guide points (which have higher weighting) to locally deform the surfaces for better fitting. Our automated approach does not introduce any other high weight guide points in addition to the contour points extracted from U-Net results, and the T-Net introduces a further improvement of 3D consistency.

## 2.3 Segmentation Pipeline

Our complete segmentation method is shown in Figure 4. It starts by inputting a stack of SAX images and two LAX images individually to the I-Net, and the predicted results ($PRDi$ in Figure 4 with two layers concatenated in the third dimension) from all slices are collected and prepared for GPM. The output results of I-Net are first thresholded at 0.5 and the coordinates of the border pixels of the resultant binary images are transformed to the desired format and inputted to GPM. GPM then produces two LV surfaces ($Mi$), which are then intersected with the image planes to give the 2D segmentation results for each slice ($SEGi$) with the same shape of I-Net output. The initial 2D segmentation results are then transformed to other slices by calculating the corresponding intersection lines using the spatial 3D coordinates provided in DICOM headers. This provides multiple transformed segmentation lines for each slice: typically, two for each SAX slice (one from each of the LAX slices) and more for each LAX slice (one from the other LAX, and one from each of the SAX slices in the stack.) For each slice all these transformed segmentations are combined to form the two additional input layers of T-Net that are introduced together with the image intensity layer. The output of T-Net ($PRDt$) has the same shape of the I-Net output, and is followed by GPM producing 3D ($Mt$) and 2D ($SEGt$) segmentation.

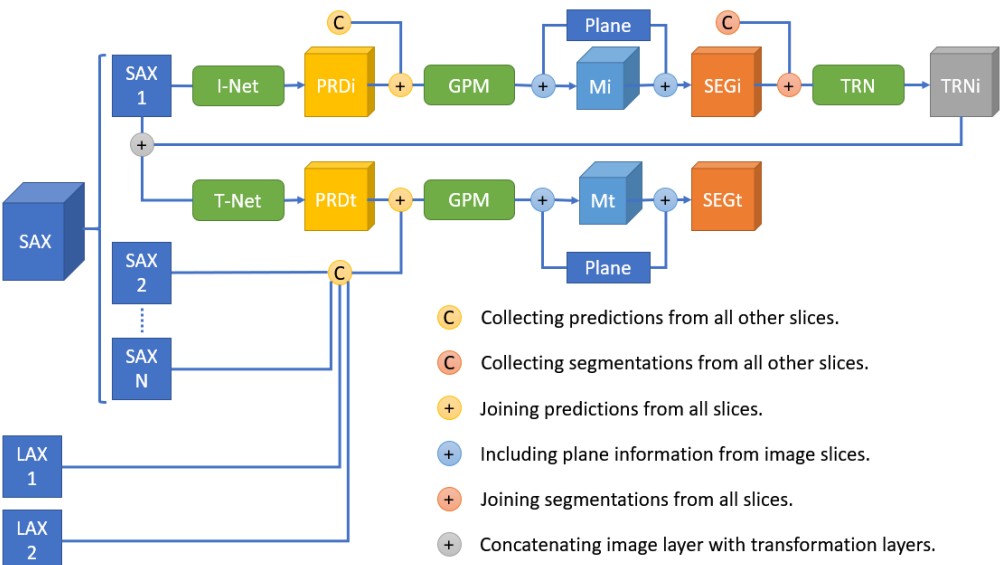

Figure 4: Overall segmentation pipeline.

# 3 Datasets and Experiments

## 3.1 Datasets

Imaging was carried out on a preclinical $9.4\,T$ MR system (Varian/Agilent, Santa Clara, US) on six male $C57Bl6/J$ mice (body weight $26.2 \pm 1.1g$), for which two repeated scans were performed within 3 days, yielding 12 scans in total. Each scan consisted of 8 to 9 SAX slices and 2 LAX slices, and for each slice two different resolution images were scanned [21]. Due to the similarity of anatomy in the same slice, the high-resolution image was down-sampled to the low resolution and pairs of scans from the same mouse were grouped together to avoid one of them being used for training and the other for testing. Each scan contained multiple time frames covering the most informative part of cardiac cycle, including End-Diastole (ED) to End-Systole (ES) frames. We used all frames to

expand our training data, thus forming 24 datasets. All 24 datasets were analysed manually, with the GPM models generated using the manual procedure used as reference.

## 3.2 Training Network

We used a leave-one-out protocol on 12 different scans, taking out one scan from the 11 training datasets for model selection. We used randomly initiated weights for both I-Net and T-Net. The transformation layers for T-Net are created using references from other slices. Due to the limited amount of data, massive data augmentations were applied, including rigid transformation, rotation, mirroring, scaling and intensity variation, and for each slice, more than 5000 variations are used on average. We used cross-entropy as loss function, and Adam as optimiser with a reducing learning rate (10 epochs for each learning rate $10^{-3}$, $10^{-4}$ and $10^{-5}$ subsequently).

# 4 Results and Discussion

## 4.1 U-Net Results

One of our key contribution of this paper is proposing a deep learning segmentation method that fuses features from intersecting slices in a U-Net and improving the 3D consistency of the output results. In this section, we evaluated the inconsistency between intersecting slices of the predictions for both I-Net and T-Net.

The amount of inconsistency between contours of two intersecting slices are defined as the distance between two contours along the intersecting line in this paper. By taking advantage of our existing segmentation algorithm, we constructed transformation layers using binary masks generated by thresholding the predictions of the networks at 0.5. For each slice, we calculated the distance between the tips of the transformation layers and the boarder of the binary mask along the intersecting line.

The average distance for I-Net result is 170 $\mu m$, and is 119 $\mu m$ for the T-Net result. By introducing transformation layers and combining information from intersecting slices, the amount of discrepancy between intersecting contours was reduced by 30% on average.

## 4.2 2D Segmentation Results

We also calculated the dice similarity coefficients for LV cavity and LV myocardium between 2D segmentation results (initial results $SEGi$ and final results $SEGt$) and the 2D reference acquired by intersecting the reference 3D surface and the image plane, similar to the process of producing segmentation results from automatically generated GPM surfaces. Each frame is considered as one case, and the results are averaged based on individual cases. For I-Net, the dice coefficient has an average value of 0.89 for LV cavity and 0.84 for LV myocardium. For I-Net, the dice coefficient has an average value of 0.88 for LV cavity and 0.83 for LV myocardium.

The results are almost identical, giving an impression that spatially consistent contours are not improving the segmentation results. However, since dice coefficient is highly sensitive to transformations, and a rigid transformation of one pixel reduces the value dramatically but has very little affects on cardiac function assessment.For this reason, we further compared the initial and final 3D segmentation results in cardiac function estimation.

## 4.3 Cardiac Function Results

We evaluated four standard metrics for cardiac function, including ED Volume (EDV) of the LV cavity, ES Volume (ESV) of the LV cavity, Ejection Fraction (EF), and LV Mass (LVM) of myocardium.

For each frame, our segmentation method generates two sets of GPM surfaces ($Mi$ and $Mt$ in Figure 4), and for each of them we calculate the LV cavity volume and the LV myocardium mass. The cavity volume is used to identify two key frames: ED and ES frames, where ED frame is the most relaxed state of the heart which has the largest cavity volume, and ES frame is the most contractive state of the heart which has the smallest cavity volume. Since the cine scan might not include a complete cardiac cycle, we find the frame with the smallest cavity volume (ES frame) first and then identify the frame with the largest cavity volume before the ES frame as ED frame, as we know that ED frame is

always taken first during acquisition in our database. We then calculate Ejection Fraction:

$$EF = \frac{EDV - ESV}{EDV} \times 100\%$$

. LVM is calculated by averaging values calculated for all available frames.

The initial segmentation results $Mi$ compared with our manually generated GPM results have an average error of $0.8 \pm 9.0\ mg$ in LVM, $-2.6 \pm 2.9\ \mu l$ in EDV, $2.5 \pm 2.8\ \mu l$ in ESV, $-5.6 \pm 3.8\ \%$ in EF. The final segmentation results $Mt$ comparing with our manually corrected GPM results have an average error of $-3.0 \pm 7.1\ mg$ in LVM, $-1.7 \pm 2.6\ \mu l$ in EDV, $2.6 \pm 2.9\ \mu l$ in ESV, $-5.0 \pm 3.8\ \%$ in EF. (Mean difference $\pm$ SD of differences, n = 24) The corresponding Bland-Altman plots are shown in Figure 5.

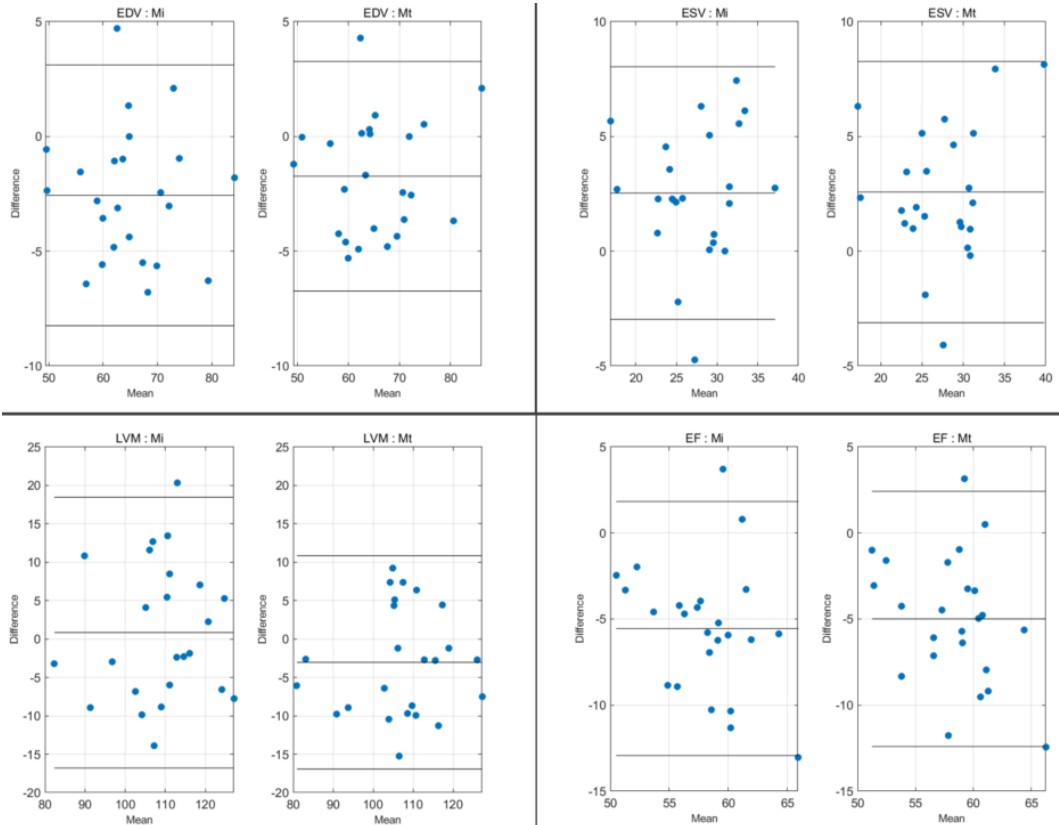

Figure 5: Bland-Altman plots of cardiac function metrics.

From the cardiac function results, we see general improvements of LVM and EDV when the consistency of 2D contours is improved using the T-Net. For the other metrics that is not as clear. Initial and final segmentation results are both similar to the references which are calculated from manually corrected GPM models, and including T-Net gives better overall performance.

## 5 Conclusion

In this paper, we propose a segmentation method that takes a stack of SAX images and two LAX images of mice cine MR scans and produces both 2D and 3D segmentation results, which are then used for key frame identification and cardiac function assessment. We also propose an efficient and effective 2D U-Net, which improves the spatial consistency by fusing information from different slices, and producing better quality GPM surfaces. The GPM surfaces are then used to estimate cardiac function metrics, which are close to the manually generated and evaluated GPM models. The entire segmentation approach is fully automated, without the requirement of any human effort. Assisted by parallel computing and GPU computing, a complete analysis of a cine-scan only take a

few minutes. Our deep learning method is designed for small database, however, the performance is expected to be further improved with an increase on the number of dataset.

The proposed deep learning segmentation approach has great potential as it takes any number of input slices (unlike typical pure neural network method which takes fixed size input and output), and begins to show its advantage when slices intersect with each other. It also gives more flexibility for data acquisition, and it is possible to use fewer intersecting slices for LV segmentation. We see better cardiac function assessment from the correction of spatial inconsistency of the contours, however, it is possible that other properties of the contours have more impact for GPM method; this will be studied in future work.

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
