# OpenReview forum: "Fully Automated Segmentation of the Left Ventricle in Small Animal Cardiac MRI"
_MIDL.amsterdam/2018/Conference — Submitted to MIDL 2018_

### Review · AnonReviewer1 · 2018-05-08
**Interesting concept for combined long-axis and short-axis segmentation**

**Rating:** 3
**Confidence:** 2

**Review:**

The concept proposed for combined segmentation of long-axis (LA) and short-axis (SA) cine CMR images is very interesting. It is always difficult to decide which SA images to include in the segmentation and ventricular volume quantification (slices between the valve plane and apex). Being able to register the LA and SA images allows for more accurately defining the valve plane and apex slice in the SA data and will improve the segmentation and volume calculation accuracy.

The paper is written in a rather unclear manner, with some minor language mistakes (e.g. "boarder of the binary mask"). My "major" point is that Figure 4 and associated description should be improved. It is - in the figure - very hard to see what is "input" and "output". I advise to use lines with arrows at the end to see from where to where data is flowing.

Furthermore, the evaluation of the estimation of the cardiac function metrics is not very convincing that the proposed method offers significant improvement over the baseline.


**Special Issue:**

No

---

### Review · AnonReviewer2 · 2018-05-08
**Segmentation method using long and short axes MR images with 2D U-nets and Guide Point Modelling, described unclearly with no baseline method to compare to.**

**Rating:** 3
**Confidence:** 2

**Review:**


Summary

A segmentation method comprising of two 2D U-nets, each using  short axis (SAX) and long axis (LAX) images, and fusing their predictions to generate 2D segmentations of left ventricle (LV) from mice cardiac MRI .  3D surfaces are reconstructed from 2D segmentations using guide-point modelling (GPM). A small dataset is used with extensive data augmentation for evaluation and the results are presented in comparison with manually generated references using GPM with no baseline methods to compare with.

Pros

1. Use of two 2D U-nets, each operating on the SAX and LAX images, and fusing their predictions at the  intersecting slices is a novel idea.
2. The method is presented to work well with small datasets, which can be useful even in other medical imaging applications.
3. Evaluations are performed both with Dice similarity and cardiac function  measures, giving an insight into the segmentation accuracy and clinical usefulness.

Cons

1. Description of the method is either incomplete or unclear. For example, in Section 2.1 it is stated that “additional two layers are the transformation layers as explained later in this section”. However, there was no clear description of what these transformation layers were. This makes the role of T-net unclear.
Further, the segmentation pipeline described in Section 2.3 is ambiguous. For instance, blocks TRN and TRNi are not explained.
2. In Section 3.2, it is reported that data augmentation is performed due to limited amount of data, stating “more than 5000 variations are used” for each slice. This is an extreme extent of augmentation. Exact nature of these transformations and their influence on training the model must be discussed. Is it reasonable to make upto 5000 variations of slices to train the model while it is also reported in Section 4.2 that “dice coefficient is highly sensitive to transformations”?
3. As there is no baseline method to compare against, the reported performance measures do not clearly quantify the performance of the proposed segmentation method.


**Special Issue:**

No

---

### Review · AnonReviewer3 · 2018-05-09
**An efficient 3D segmentation method is proposed for small animal cardiac MR images. A combination of long-axis and short-axis images is used, by combining convolutional neural networks and a guide-point modelling method. Due to small dataset used thus far and the presented results, we view the paper as preliminary.**

**Rating:** 2
**Confidence:** 2

**Review:**

In this paper, an efficient 3D segmentation method is proposed for small animal cardiac MR images. A combination of long-axis and short-axis images is used, by combining convolutional neural networks
and a guide-point modelling method. A small training set is shown to give accurate results. The segmentation method is shown to take a stack of SAX images and two LAX images of mice cine MR scans and produce both 2D and 3D segmentation results, which are then used for key frame identification and cardiac function assessment.

Authors present a method for segmentation of LV using a SAX and LAX MR images of the LV. The method relies on two U-nets that are used to extract contour point coordinates later fed to a GPM (Guide-Point Modelling algorithm) which is a clinically accepted method for the generation of the LV shape based on a 3D finite-element model.
In measuring segmentation results often Dice or Jaccard similarity metrics are utilized. Here the authors used average Dice for 2D slices. Dice results are not clear as result of a typo. Moreover, the Dice results lack standard deviation making interpretation problematic. May be relevant to use 3D Dice coefficient for validation.
Section 4.1 the "average distance" should be better explained. The author mentions that their method improves this measure, but it is not clear what it means. Not sure what this sentence means: "the distance between the tips of the transformation layers and the boarder of the binary mask along the intersecting line". If this evaluation measure is custom made by the author it should be properly defined, otherwise a relevant reference paper should be cited.
The focus on mice , the small amount of data used for training and the lack of comparison to state of the art methods (e.g https://arxiv.org/pdf/1709.04496.pdf “An Exploration of 2D and 3D Deep Learning Techniques for Cardiac MR Image Segmentation”) reduces the  applicability of the work to the deep learning community.
Results:  Using DICE, it seems that the additional T-Net does not improve the results. When using EF for performance measure it seems that the additional T-Net does not increase significantly the performance (and decrease in some cases). No p-value is provided. Based on the above, the following statement: "and including T-Net gives better overall performance" –is not supported by the results provided in this paper.

Strengths
-Automating a GPM algorithm which currently requires user interaction.
-The process can receive multiple LAX and SAX slices
-The idea of using Transformation layers for the T-net is interesting and can serve a basis for future experimentations

Weaknesses
-Comparison with 2D and 3D segmentation method was not performed- we refer to https://arxiv.org/pdf/1709.04496.pdf “An Exploration of 2D and 3D Deep Learning Techniques for Cardiac MR Image Segmentation”
-The focus on mice , the small amount of data used for training and the lack of comparison to state of the art methods (e.g https://arxiv.org/pdf/1709.04496.pdf “An Exploration of 2D and 3D Deep Learning Techniques for Cardiac MR Image Segmentation”) reduces the  applicability of the work to the deep learning community
-Comparison with non GPM based method may be of interest

Minor comments:
Figure 3: LAX & SAX labels are switched.
4.2: “For I-Net, the dice coefficient has an
average value of 0:89 for LV cavity and 0:84 for LV myocardium. For I-Net, the dice coefficient has
an average value of 0:88 for LV cavity and 0:83 for LV myocardium.”
I-net Result is mentioned twice




**Special Issue:**

No

---

### Decision · Program_Chairs · 2018-05-15
**Paper16 Acceptance Decision**

Reject